# Brain–Bone Crosstalk in a Murine Polytrauma Model Promotes Bone Remodeling but Impairs Neuromotor Recovery and Anxiety-Related Behavior

**DOI:** 10.3390/biomedicines12071399

**Published:** 2024-06-24

**Authors:** Katharina Ritter, Markus Baalmann, Christopher Dolderer, Ulrike Ritz, Michael K. E. Schäfer

**Affiliations:** 1Department of Anesthesiology, University Medical Center of the Johannes Gutenberg-University Mainz, Langenbeckstrasse 1, 55131 Mainz, Germany; katharina.ritter@unimedizin-mainz.de (K.R.); m.baalmann@uke.de (M.B.); 2Department of Orthopedics and Traumatology, University Medical Centre of the Johannes Gutenberg-University Mainz, Langenbeckstrasse 1, 55131 Mainz, Germany; christopher.dolderer@gmail.com (C.D.); ritz@uni-mainz.de (U.R.)

**Keywords:** traumatic brain injury, bone fracture, osteopathology, neuropathology, neuroinflammation, behavior, mice

## Abstract

Traumatic brain injury (TBI) and long bone fractures are a common injury pattern in polytrauma patients and modulate each other’s healing process. As only a limited number of studies have investigated both traumatic sites, we tested the hypothesis that brain–bone polytrauma mutually impacts neuro- and osteopathological outcomes. Adult female C57BL/6N mice were subjected to controlled cortical impact (CCI), and/or osteosynthetic stabilized femoral fracture (FF), or sham surgery. Neuromotor and behavioral impairments were assessed by neurological severity score, open field test, rotarod test, and elevated plus maze test. Brain and bone tissues were processed 42 days after trauma. CCI+FF polytrauma mice had increased bone formation as compared to FF mice and increased mRNA expression of bone sialoprotein (BSP). Bone fractures did not aggravate neuropathology or neuroinflammation assessed by cerebral lesion size, hippocampal integrity, astrocyte and microglia activation, and gene expression. Behavioral assessments demonstrated an overall impaired recovery of neuromotor function and persistent abnormalities in anxiety-related behavior in polytrauma mice. This study shows enhanced bone healing, impaired neuromotor recovery and anxiety-like behavior in a brain–bone polytrauma model. However, bone fractures did not aggravate TBI-evoked neuropathology, suggesting the existence of outcome-relevant mechanisms independent of the extent of brain structural damage and neuroinflammation.

## 1. Introduction

Traumatic brain injury (TBI) is a leading cause of death and disability, and sixty-nine million individuals are estimated to suffer TBI each year [1]. Unfortunately, there is no gold standard treatment for TBI besides surgical interventions and symptomatic relief [2]. In polytrauma patients, TBI with concomitant fractures of long bones represents a common injury pattern [3], and approximately one third of patients with a femoral fracture also present with head or neck injuries [4,5]. Experimental as well as clinical data provide strong evidence for reciprocal interactions between brain and bone at both traumatic spots, each influencing recovery and outcome of the other [6,7,8,9]. Therefore, investigating the pathophysiological mechanisms of this crosstalk may help to discover novel pharmacologic targets and treatment strategies [10].

Since opposing outcome-relevant effects on brain and bone have been reported, it seems reasonable to divide the reciprocal mechanisms into efferent brain-to-bone and afferent bone-to-brain mechanisms [10].

Clinical observations indicated that brain–bone crosstalk, i.e., mediated by humoral factors, leads to increased bone formation and accelerated bone healing [11]. This phenomenon has been associated with increased systemic levels of markers for hematoma formation and an increased pro-inflammatory state in TBI patients with concomitant tibial fracture [12]. In addition, osteoinductive effects have been observed in cerebrospinal fluid from TBI patients [13]. Larger callus volumes in TBI patients with tibial or femoral fracture are also linked to increased serum levels of cytokines, growth factors, and hormones at 4 weeks [14] or 5 weeks after polytrauma [15], respectively. Additionally, increased numbers of M2 macrophages in the callus tissue [16] and TBI-induced adrenergic signaling promoting bone marrow myelopoiesis [17] have been reported. Animal studies expanded the clinical observations of bone healing and remodeling after fractures and provided evidence for central regulation of bone remodeling [18]. Among the potential factors associated with these processes are circulating growth factors, chemokines, and hormones [19,20,21] including leptin [22] and norepinephrine [23] as well as amino acids and lipid metabolites including leucine and arachidonic acid, respectively [24,25]. Moreover, small extracellular vesicles released by injured neurons have been shown to promote osteoblastic differentiation after tibial fracture via specific micro RNAs [26] suggesting direct brain-to-bone signaling after TBI. Taken together, clinical and experimental studies have shown that brain-to-bone crosstalk promotes post-fracture bone growth and remodeling after TBI. 

Conversely, bone-to-brain crosstalk has been associated with increased mortality, worse outcome at the time of discharge, and higher rates of neurosurgical interventions in TBI patients with femoral fractures [27]. Secondary brain damage, characterized by cerebral edema, ischemia, and neuroinflammation in the early stage of TBI, and astroglial scar formation and neurodegeneration in the late stage of TBI, is exacerbated by the concomitant presence of a long bone fracture [28,29,30]. Increased release of pro-inflammatory cytokines such as interleukin-1 beta (IL-1β) or tumor necrosis factor alpha (TNFα) is associated with elevated edema and blood brain barrier (BBB) disruption after 24 h and increased expression of glial fibrillary acidic protein (GFAP) and MRI diffusion abnormalities after 35 days in animals with combined TBI and tibial fracture along with behavioral impairments [29]. Elevated plasma levels of the cytokine osteopontin (OPN) have also been suggested to be associated with aggravated brain histopathology in combined TBI and femoral fracture at 5 days after injury [31]. 

However, most studies focused on either brain injury or bone fracture and consider the other merely a necessary co-factor. 

In this study, we investigated the consequences of long-term reciprocal brain–bone interactions and assessed the course of behavioral impairments. To this end, mice were subjected to TBI and/or femoral fracture (FF) or sham surgery. Brain and bone tissues were processed at 42 days after trauma, and behavioral assessments were conducted over 42 days.

## 2. Materials and Methods

Animals and study groups. Adult female C57BL/6N mice (Janvier Labs, Le Genest-Saint-Isle, France), 8–9 weeks old at the time of injury, were housed under standard conditions (22–24 °C, 12 h light/dark cycle, 55% humidity, food and water ad libitum). Animal experiments were carried out in accordance with ARRIVE guidelines and approved by the responsible governmental authority (Landesuntersuchungsamt Rheinland-Pfalz, reference number 23177-0/G17-1-062). 

Five days before beginning of the experiment, mice were transferred to the laboratory facilities and randomized to four different study groups. Two groups were subjected to isolated trauma of controlled cortical impact (CCI, n = 15) or fracture of the left femur (FF, n = 15), the third group received combined injury (CCI+FF, n = 15), while the fourth group was conducted as sham (sham, n = 12). An overview of the experimental design and timeline is shown as Figure 1:

Behavioral assessment. Mice were examined for behavioral changes using various tests commonly used to assess of neurological, motor, exploratory, and anxiety-like phenotypes. We used a modified Neurological Severity Score (NSS) developed from Tsenter et al. [32], in which mice received scores of 0–13 (higher scores indicating greater impairment). The NSS included assessment of general behavior, coordination and balancing skills, and overall motoric impairment and was conducted one day before as well as 1, 3, 7, 14, 21, and 42 days post injury (dpi). As neurological assessment relies in major parts on motoric function a leg performance test (LPT, 0–6 points, higher scores indicating greater impairment) was additionally implemented, focusing on strength and mobility of the lower limbs in detail [31]. 

In addition, each scoring episode included a 3 min video-tracked (Ethovision XT 14, Noldus Information Technology BV, Wageningen, The Netherlands) open field test (OFT) in a 40 × 40 cm arena with automatic registration of mean velocity in movement and total distance travelled within each sequence to gain an impression of mice’s general locomotor abilities. Time spent in the center and border zone of the arena as well as duration needed to exit from the central zone at the beginning of the recording episode were separately analyzed as indicators of deranged anxiety-related behavior. Motoric impairments were further addressed by rotarod (RR) test at 21 dpi, a post-traumatic time point showing impaired RR performance in mice after CCI [33]. Mice absolved two training sessions before performing the actual trial and the speed of the rotating wheel was increased constantly from 4 to 40 revolutions per minute over a time interval of 300 s. To extent testing of behavioral dysfunction, an elevated plus maze (EPM) was conducted at 20 dpi. Mice were placed in the center of the EPM arena (30 × 5 cm per arm, 15 cm wall height, 40 cm above floor) and video-tracked for 5 min. Time spent in open and closed arms and frequency of entries were analyzed, as well as the mean velocity and the total distance travelled by each animal. 

Analgesia and surgical procedures. Perioperative analgesia was ensured by tramaldol (100 mg/mL to the drinking water, Ratiopharm, Ulm, Germany) provided from two days before surgery until 7 dpi. In addition, administration of fentanyl (50 μg/kg, intraperitoneal, Janssen-Cilag NV, Beerse, Belgium) was administered 15 min before anesthesia induction (isoflurane 4 vol% for 60 s) followed by 1.5 vol% for anesthesia maintenance. 

Reflexes were tested repeatedly to verify adequate depth of anesthesia, and spontaneous breathing was upheld for the entire procedure. Body temperature was measured continuously via rectal probe and preserved at 37 °C using a heating pad (Thermolux, Murrhardt, Germany) during the intervention.

CCI and femoral fracture were induced as previously described [31]. Briefly, mice were fixed in a stereotactic frame and CCI was induced to the right parietal cortex using a Benchmark™ Stereotaxic Impactor (Leica Biosystems, Wetzlar, Germany; impactor tip diameter: 3 mm, impact velocity: 6 m/s, impact duration: 200 ms, displacement: 1.5 mm). Bone fracture was induced to the left femur using a three-point fracture device after insertion of the MouseScrew^®^ (both RISystem AG, Davos, Switzerland). Bone fracture was confirmed and documented by radiography (35 kV/5 s) using an animal X-ray device (Faxitron MX-20, Faxitron Bioptics, LLC, Tucson, AZ, USA).

After completion of surgical procedures, mice were transferred into their cages and placed into a neonatal incubator (Babytherm 8000, Draeger, Luebeck, Germany) for two hours and then returned to their home cages. The sham procedure included analgesia, anesthesia, and skin incisions as described above without further surgical intervention; management and duration of anesthesia as well as perioperative parameters were upheld identical in all groups. Mice were screened for early termination criteria-predefined as ≥20% loss of initial body weight and/or exhibiting signs of severe pain or discomfort in accordance with national and international recommendations—several times daily. Overall mortality rate was 1.75% (one CCI+FF mouse deceased 14 dpi), and no animals fulfilled the early termination criteria.

Brain histology and immunofluorescence staining. Mice were killed by decapitation in deep anesthesia using 4 vol% isoflurane at 42 dpi. Brains were carefully dissected, frozen in powdered dry ice, and stored at −20 °C until sectioning using a cryotome (Cryo-Star NX70, Thermo Fisher Scientific). Brain sections, 12 µm thick, were collected on Superfrost^®^ Plus Slides (Thermo Fisher Scientific Inc., Waltham, MA, USA) and processed for cresyl violet staining and lesion volumetry as described [31]. 

Areas with an absence of cresyl violet staining from 16 sections (from Bregma +3.14 mm to −4.36 mm) were added and multiplied by 500 μm, resulting in the corresponding volumes. Width of the granular cell layer (GCL) of the suprapyramidal blade of dentate gyrus was determined in three predefined locations in the ipsi- and contralateral hemispheres, and mean values were calculated from two sections (Bregma −1.8 to −2.0 mm) of each animal [31].

Immunofluorescence staining was performed essentially as described [31]. Briefly, sections were post-fixated using 4% paraformaldehyde in phosphate-buffered saline (PBS), incubated for 1 h in blocking solution (5% normal goat serum, 0.5% bovine serum, and 0.1% Triton-X100 in PBS), followed by overnight incubation with specific primary antibodies (rat anti-GFAP Thermo Fisher Scientific Cat# 13-0300, 1:500 and rabbit anti-Iba-1 WAKO Cat# 019-19741, 1:1000) and appropriate secondary fluorophore-conjugated antibodies (goat anti-rat IgG, Alexa Fluor 488, Thermo Fisher Scientific, Cat# A-11006; goat anti-rabbit IgG, Alexa Fluor 568 Thermo Fisher Scientific, Cat# A-11011, both diluted 1:500 in blocking solution) and mounted in ImmunoMount (Thermo Fisher). Images from two brain sections (Bregma −1.86 mm to −2.86 mm) from each animal were acquired by confocal laser microscopy (LSM510, 20x objective, Zeiss) using identical settings. An investigator blind to the study groups analyzed anti-GFAP and anti-Iba-1 immunofluorescence essentially as described [34] using ImageJ software (NIH Image, https://imagej.net/) and the “Analyze Particle” plugin.

Bone histology. After euthanasia, the left femur from each mouse was removed in toto, and half of the collected samples (sham n = 5, CCI n = 5, FF n = 6, CCI+FF n = 5) were processed for histological analyzes essentially as described [35]. Briefly, femora were decalcified, embedded in paraffin, cut to 5 µm sections, and stained with hematoxylin and eosin (HE). Microscopic images were acquired (Keyence BZ-X800, Neu-Isenburg, Germany); the areas of cortical bone and the medullar cavity were outlined by freehand selections using ImageJ and compared between groups. 

Gene expression analysis. To quantify gene expression in brain tissue samples, RNA extraction, cDNA synthesis, and qRT-PCR was performed as previously described [36]. Briefly, RNA was extracted from coronal brain tissue sections (Bregma +0.64 mm to −2.86 mm) and transcribed to cDNA, and oligonucleotide primers genes were used to amplify specific gene target sequences (Table 1) using a Light Cycler^®^ (Roche Molecular Systems Inc., Pleasanton, CA, USA) and SYBR Green Mix Plus ROX Vial (Thermo Fisher Scientific) or specific oligonucleotide probes LightCycler^®^ 480 Probes Master (Roche Molecular Systems Inc., Pleasanton, CA, USA). Samples were analyzed in duplicate, and quantification was performed using a target specific standard curve and normalization to the reference gene cyclophilin A (Ppia) [37]. 

To quantify gene expression in bones, 50 mg of femoral bone tissue was used and processed for RNA extraction, cDNA synthesis, and qRT-PCR as described [31]. Specific oligonucleotide primers (Table 2) and the SYBR Green (PowerUp™ SYBR^®^ green master mix, Applied Biosystems, Foster City, CA, USA) were used for target gene sequence amplification with the qTower3 cycler (Jena Analytik, Jena, Germany). The 2^−ΔΔCt^ method was used to calculate gene expression values in bones from experimental injury groups relative to the sham group [31].

Statistical analysis. GraphPad Prism (version 8, GraphPad Software Inc., San Diego, CA, USA) was used to analyze data. Rout’s test was used to identify outliers; the Shapiro–Wilk normality test and QQ plots were used to determine the (non-) parametric distribution of data. Depending on data distribution Student’s *t*-test or Mann–Whitney U test were used to calculate p values in pairwise comparisons, while multiple groups were analyzed by ordinary one-way analysis of variance (ANOVA) or Kruskal-Wallis test and Holm–Šidák or Dunn’s post hoc test depending on (non-) parametric distribution of data. Two-way ANOVA followed by Holm–Šidák or Tukey’s post hoc test were used to analyze differences between groups in body weight or behavioral tasks at multiple time points. Data are presented as mean ± standard error of the mean (SEM); * *p* < 0.05, ** *p* < 0.01, and *** *p* < 0.001.

## 3. Results

### 3.1. TBI Up-Regulates BSP Gene Expression in the Fractured Bone and Promotes Cortical Bone Formation

X-ray images from femoral fractures were taken at consecutive time points after the surgical intervention to verify adequate fracture induction and osteosynthetic stabilization, callus formation, and fracture healing over the 42-day study period (Figure 2A). 

First, we investigated efferent effects of brain-to-bone crosstalk, as clinical and experimental studies have reported enhanced bone healing processes associated with TBI [7,8]. Gene expression associated with inflammation, bone metabolism, and repair [31] was analyzed by qPCR in femoral bone tissue (Figure 2B). While Bdnf, Grn, Mrc1, Bglap, Spp1, and Runx2 mRNA expression was not significantly altered among CCI, FF, and CCI+FF mice. Alp expression was increased in animals with femoral trauma compared to isolated CCI (FF: 3.72 ± 0.59; CCI: 1.45 ± 0.26; mean ± SEM, *p* = 0.0230) and tended to be reduced in the fractured bone by CCI (CCI+FF: 3.12 ± 0.63; CCI+FF vs. FF: *p* = 0.0778). Remarkably, BSP gene expression (encoded by Ibsp) was significantly increased in mice with combined trauma, CCI+FF as compared to isolated CCI (CCI+FF: 3.29 ± 0.39; CCI: 1.53 ± 0.36; mean ± SEM, *p* = 0.0080), and FF (1.70 ± 0.27; mean ± SEM, CCI+FF vs. FF *p* = 0.0111). 

HE-staining of femoral bone at 42 dpi revealed a relative increase of cortical bone (CCI+FF: 46.42 ± 2.97; FF: 36.79 ± 2.63, mean ± SEM, *p* = 0.0369) and a relative decrease of medullar cavity (CCI+FF: 53.58 ± 2.97; FF: 63.21 ± 2.63, mean ± SEM, *p* = 0.0369) in bone tissue sections of CCI+FF mice in comparison to isolated FF, as well as an increased ratio of cortical bone/medullar cavity (CCI+FF: 0.89 ± 0.11; FF: 0.59 ± 0.058; mean ± SEM, *p* = 0.0321) (Figure 2C).

These results indicated that TBI up-regulates BSP gene expression in the fractured bone and promotes cortical bone formation.

#### 3.1.1. Femoral Fracture Does Not Aggravate Long-Term Brain Histopathology after TBI

At 42 dpi, brain histopathology was examined using volumetry of brain hemispheres and the hippocampal granule cell layer (GCL) width was determined (Figure 3), a brain region affected by secondary injury in the CCI model of TBI [38]. CCI produced substantial tissue loss in the ipsilesional hemispheres, but the volume of the remaining brain tissue in the lesioned hemispheres was comparable between the two groups with cerebral trauma (CCI: 131.4 ± 2.04; CCI+FF: 135.8 ± 2.45, mean ± SEM, CCI vs. CCI+FF: *p* = 0.3124). GCL width was reduced in the lesioned hemispheres (CCI: 46.80 ± 2.62; CCI+FF 46.85 ± 1.97, mean ± SEM) compared to sham (52.96 ± 1.59; mean ± SEM, CCI vs. sham: *p* = 0.0432; CCI+FF vs. sham: *p* = 0.0432) and FF (54.85 ± 1.45; mean ± SEM, CCI vs. FF: *p* = 0.0207; CCI+FF vs. FF: *p* = 0.0246), as expected, but not different between the groups subjected to CCI or CCI+FF (CCI vs. CCI+FF: *p* = 0.9873) (Figure 3A). 

CCI leads to long-lasting activation of astrocytes and microglia in TBI patients [28] and the CCI model of TBI [33]. Immunohistochemistry using specific antibodies to GFAP or Iba1 was used to examine astroglial and microglial reactivity in the perilesional cortex. Both groups subjected to CCI showed increased numbers of GFAP- (CCI: 58 ± 7.84; CCI+FF: 48.71 ± 6.58) and Iba1-immunopositive particles (CCI: 25.6 ± 2.78; CCI+FF: 26.21 ± 3.22) compared to FF (GFAP: 10.29 ± 2.28; *p* < 0.0001; Iba-1: 12.64 ± 0.94; CCI vs. FF: *p* = 0.0009; CCI+FF vs. FF: *p* = 0.0007) and sham (GFAP: 5.81 ± 2.41; *p* < 0.0001; Iba-1:12.18 ± 0.60; CCI vs. sham: *p* = 0.0009; CCI+FF vs. sham: *p* = 0.0009), as expected, but no remarkable difference was noted between both groups with cerebral trauma (CCI vs. CCI+FF: *p* = 0.4327 [GFAP]; *p* = 0.9771 [Iba1]) (Figure 3B).

These results indicate that femoral fracture does not aggravate long-term brain histopathology in the CCI model of TBI.

#### 3.1.2. Femoral Fracture Has no Influence on Late TBI-Induced Gene Expression of Inflammation-Associated Markers

Gene expression in the perilesional brain tissue of the lesioned hemisphere was analyzed by qPCR using a panel of inflammation-associated markers predominantly expressed by astrocytes or microglia (Figure 4). We observed highly significantly increased expression of all marker genes at 42 dpi in CCI or CCI+FF mice as compared to sham or FF, respectively (Tspo, Gfap, Cd74, C3, Serpa3n, Spp, all *p* < 0.001). However, femoral fracture had no influence on late TBI-induced gene expression of inflammation-associated markers.

#### 3.1.3. Concomitant TBI and Femoral Fracture Prolongs Recovery and Affects Anxiety-Related Behavioral Patterns

Mice of all groups except FF suffered from a significant weight loss at 1 dpi compared to their pre-traumatic status (*p* < 0.5), but regained their body weight quickly until 3 dpi, and body weight did not vary significantly between the study groups at any time point (Figure 5A). A series of tests was performed to assess neuromotor function and anxiety-related behavior comprising a neurological severity score (NSS), leg performance test (LPT), rotarod performance test (RR), open field test (OFT), including exit from center (EFC) as a subvariable, and the elevated plus maze (EPM). 

First, data analysis of the NSS (repeated assessment with higher scores reflecting increased impairment) revealed that mice with CCI as a single injury showed an increased NSS compared to sham until 3 dpi (Figure 5B, CCI: 3.90 ± 0.70, sham: 0.75 ± 0.30, mean ± SEM, *p* = 0.0033), while no statistically significant differences were noted from 7 dpi until study’s endpoint. Comparing the NSS of CCI mice to their pre-traumatic (preop.) status, the CCI-induced impairment remained statistically significant until 7 dpi (CCI-preop: 0.40 ± 0.23; CCI-7dpi: 3.33 ± 0.80; mean ± SEM, *p* = 0.0121) and declined until 42 dpi (CCI-42dpi: 2.36 ± 0.62, mean ± SEM, preop vs. CCI, *p* = 0.2485), indicating a full recovery of CCI mice in terms of NSS assessment.

In contrast, CCI+FF mice subjected to combined injury showed an increased NSS compared to all other groups up to 14 dpi (Figure 5B) (CCI+FF vs. sham: *p* = 0.0089; CCI+FF vs. FF: *p* = 0.0073; CCI+FF vs. CCI: *p* = 0.0089). In comparison to sham, the NSS of CCI+FF mice was increased until the endpoint of this study at 42 dpi (CCI+FF: 4.39 ± 0.84; sham: 1.75 ± 0.62; mean ± SEM, *p* = 0.0.262). Regarding recovery over time, NSS remained significantly higher in CCI+FF mice compared to their pre-traumatic status until 42 dpi (pre-OP: 0.32 ± 0.21; 42 dpi: 4.39 ± 0.84, mean ± SEM, *p* = 0.0001). However, we also observed increased NSS in mice with isolated FF in comparison to sham until 7 dpi (FF: 4.66 ± 0.54; sham: 2.25 ± 0.76, mean ± SEM, *p* = 0.0322), as the NSS relies in major parts on the motoric function of the limbs [31]. NSS of CCI mice was lower than in FF as a single injury during the experiment except for 1 dpi, again underscoring the rapid recovery after isolated CCI compared to combined injury as well as the impact of the fracture impact on neuromotor scoring.

To begin to differentiate effects caused by the fracture from actual neurological impairment, the LPT was performed (Figure 5C, repeated assessment with higher scores reflecting increased impairment). At 1 dpi, mice with combined injury showed a statistical trend towards increased impairment of the lower limbs in comparison to isolated FF (CCI+FF: 3.10 ± 0.33; FF: 2.03 ± 0.30, mean ± SEM, *p* = 0.0559), but not at later time points until 42 dpi. Accordingly, assessing neuromotor function using the RR performance test did not reveal differences between the experimental groups at 21 dpi (Figure 5D).

The results suggest that lower limb impairment is slightly increased at 1 dpi by concomitant CCI and recovers quickly in our experimental model. 

OFT and EPM were conducted to assess explorative and anxiety-like patterns. However, locomotion is an essential component of these tests. In the OFT, the total distance travelled in the arena was video-tracked during the 3 min OFT sequence of every scoring episode and analyzed (Figure 6A,B). 

Both groups with femoral trauma explored less compared to sham at 1 and 3 dpi, but this effect disappeared at 7 dpi. An increased locomotor activity was observed in CCI and CCI+FF mice from 7 dpi until 42 dpi, consistent with previous reports on hyperactivity after CCI in mice [39]. FF mice and sham mice tended to show less locomotion, possibly due to habituation effects. We next analyzed the time mice spent in the center and border zone of the OFT and the duration required to exit from the arena’s open center to the seemingly safer border zone examines similar patterns. To reduce habituation effects as potential confounder, data were normalized to sham values prior to analysis.

This analysis showed that CCI+FF mice required longer time to exit from center (EFC) compared to CCI mice until 42 dpi (Figure 6D) (CCI+FF: 0.82 ± 0.24; CCI: 0.20 ± 0.04; FF: 0.32 ± 0.07, mean ± SEM, CCI+FF vs. CCI *p* = 0.0275; CCI+FF vs. FF: *p* = 0.0580).

The EPM was performed at 20 dpi (Figure 5C). While general motion during the 5 min recording episode was comparable among all groups, we observed reduced time spent in the closed arms of the arena (sham: 151.30 ± 18.96; CCI: 81.11 ± 19.62; FF: 87.43 ± 20.81; CCI+FF: 86.32 ± 21.99, mean ± SEM; *p* < 0.1) as well as an increased frequency of open arm entries in all groups subjected to surgical trauma in comparison to sham (sham: 144.9 ± 19.51; CCI 214.9 ± 19.69; FF: 210.3 ± 20.62; CCI+FF: 210.2 ± 22.54, mean ± SEM, *p* < 0.1).

Taken together, concomitant TBI and femoral fracture prolongs recovery and affects anxiety-related behavioral patterns.

## 4. Discussion

Long bone fractures are among the most common injuries associated with TBI [3]. Both clinical observations and experimental data from animal studies indicated reciprocal brain–bone interactions [6,7,8,9]. However, most studies focused on either brain injury [40] or bone fracture [23,41] and consider the other merely a necessary co-factor.

This experimental study examined the reciprocal effects of TBI (CCI) and femoral fracture (FF) in mice over a period of six weeks after trauma.

### 4.1. Bone Formation in the Fractured Femur Is Increased by Concomitant Brain Injury

Histological analysis of bone tissue at 42 dpi revealed increased bone formation in CCI+FF mice as compared to FF mice. This observation is consistent with previous findings from clinical and experimental studies [11,12,23,26,42]. We further observed fracture-induced up-regulation of gene markers associated with bone mineralization, metabolism and remodeling, i.e., Alpl, Ibsp, and Spp1, encoding for alkaline phosphatase, BSP, and osteopontin, respectively. Among these genes, the expression of Ibsp was significantly increased in CCI+FF mice compared to FF mice. BSP, a member of small integrin-binding ligand N-linked glycoproteins (SIBLINGs), is crucial for bone tissue formation, remodeling, and repair [43,44,45,46,47]. Our results on Ibsp expression regulation suggest that BSP contributes to enhanced osteogenesis after combined trauma of TBI and femoral fracture. Based on our previous data on reduced osseous gene expression of BSP in the early polytraumatic stage at 5 dpi [31], the results indicate stage-specific expression regulation of BSP gene expression in the context of concomitant TBI. Earlier work reported BSP expression regulation by steroid hormones, i.e., glucocorticoids [48]; however, TBI causes dysfunction of the hypothalamic-pituitary axis [49] suggesting that glucocorticoid-dependent regulation of BSP is unlikely in our polytrauma model. Another possibility is that astrocytes or microglia, which were still activated at 42 dpi as demonstrated by immunohistochemistry, released circulating factors that enhanced bone remodeling and repair. This hypothesis is in line with clinical data linking increased serum levels of cytokines and growth factors at 4 weeks after polytrauma with larger callus volumes [14]. However, our gene expression data from brain samples for IL-1β suggest that acute phase proteins including IL-1β and other pro-inflammatory cytokines are unlikely to be brain-derived contributing factors in the late phase of bone repair in our polytrauma model. In contrast, the brain gene expressions of OPN and the complement factor C3 were still increased at 42 dpi in single TBI and polytrauma. Both gene products are possible candidates for brain-to-bone signaling as they are associated with bone growth and remodeling [50,51]. However, to elucidate the mechanisms of TBI-induced bone growth and remodeling, more comprehensive analyzes than those in the present study are required, focusing on longitudinal analysis and correlations of blood biomarkers as well as osteopathological and neuropathological markers.

### 4.2. Delayed Neuromotor Recovery after Combined Brain and Bone Injuries

Regarding clinical manifestation, mice with combined injury of CCI and FF displayed more pronounced behavioral deficits compared to all other groups up to 14 dpi, and up to 42 dpi compared to sham and showed a prolonged recovery period in the repeatedly assessed NSS. As neurological assessment in mice relies in major parts on motoric function, the femoral fracture must be considered a potential disruptive element. In this study, groups with FF showed no greater motoric impairment in the RR test at 21 dpi or in general mobility during the sequentially used OFT, exposing the osseous trauma as a rather insignificant confounder in testing accuracy. However, it should be noted that repeated assessment of motor function potentially generates training effects and might be considered as a rehabilitation process [52]. To reduce this effect as a potential confounder, the assessment schedule remained identical among all subjects and focused on the performance in the NSS and OFT rather than repeated rotarod testing. Neurobehavioral impairment after TBI not only includes simple motoric disabilities but also reduction of higher cognitive functions like short- and long-term memory or spatial learning [53,54]. Both OFT and the EPM serve as well-established tools to assess anxiety-related behavior in rodents [55]. EPM was performed at 20 dpi and revealed reduced anxiety-related exploring patterns in all groups with manifest trauma, and mice with combined injury required notably more time to complete the exit from circle task in the OFT until study’s endpoint at 42 dpi. In conclusion, a concomitant femoral fracture aggravates the late neuromotor deficit after TBI and leads to disordered anxiety-associated behavior. This observation is consistent with an earlier study showing that concomitant bone fracture delays cognitive recovery from TBI in mice [40].

### 4.3. Altered Anxiety-like Behavior after Combined Brain and Bone Injuries

Previous works have elucidated anxiety-like behavior as a vulnerable parameter after isolated TBI, yet findings and interpretations vary. While post-traumatic stress disorder, depression or cognitive and affective disorders are the common findings among patients [56], studies in rodents also suggest time-dependent effects linked to a late decrease of GABA-mediated inhibition in the amygdala [57,58]. Long-term deficits in spatial learning and memory function were ascribed to phenotypic changes and activation of microglia after TBI [53]. In this study, we examined activation of astroglia and microglia in the perilesional brain tissue by immunohistochemistry and qPCR, but found no evidence for their aggravation by the additional femoral trauma. Increased histological damage in the cerebral lesion itself and in adjoining regions were found in rodents subjected to TBI and long bone fracture compared to isolated TBI but could not be reproduced in this study [30,31]. 

In the present study, mice that sustained a femoral fracture as a single injury showed abnormalities in the EPM comparable to those of mice with brain and combined trauma. Furthermore, after a rapid initial recovery, they showed increasing NSS over time. Post-traumatic or postoperative cognitive dysfunction is a common problem among elderly patients suffering from hip fractures, but its pathogenesis is not fully understood [59,60]. In experimental settings, activation of the classical inflammatory TLR4/MyD88 pathway [61] or dysfunction of regulatory T cells [62] are–among others-mechanisms under investigation. In our study, gene expression analysis of several (neuro-) inflammatory markers was performed, but we did not identify a specific mediator for the observed clinical symptoms, a circumstance probably due to the late posttraumatic examination time-point.

### 4.4. Limitations of This Study

This study has some limitations to be considered. First, we considered group housing important to avoid behavioral effects due to social isolation [63]. Therefore, only female mice were studied to enable group housing and avoid additional injuries caused by ranking battles, which often occur in group housing of male mice. However, sex-specific effects have been reported in patients as well as in mice after TBI [64,65,66,67] or bone fracture [68,69]. Second, age is another outcome-relevant factor in experimental models of TBI [70,71] or bone fracture [72,73], which has not been considered in the present study using only 8–9 weeks-old mice. It would be therefore important to examine the influence of age and sex on the histopathological and behavioral outcomes in combined injury models. Third, behavioral testing using rotarod or the EPM was not performed at the end-point of this study at 42 dpi, thereby preventing conclusions on the persistence of the observed effects at 20 dpi or 21 dpi, respectively. Finally, additional tests to assess cognitive function would have provided a more comprehensive picture of changes in the behavioral spectrum in our combined injury model.

## 5. Conclusions

To date, only a few studies have examined the reciprocal interactions in experimental TBI and concomitant bone fracture. Here, we tested the hypothesis that brain–bone polytrauma mutually impacts neuro- and osteopathological outcome and provided new insights into the pathological relevance of crosstalk between both traumatic spots. Brain-to-bone crosstalk enhanced bone healing, impaired neuromotor recovery and anxiety-like behavior. Furthermore, we propose that an increased osseous BSP gene expression associates with accelerated bone healing after polytrauma. This finding might be important for further elucidating mechanisms of BSP and its regulation under pathophysiological conditions in the context of bone remodeling.

While bone-to-brain crosstalk seems to impact overall behavior and recovery, bone fracture did not aggravate TBI-evoked neuropathology and neuroinflammation. This suggest the existence of outcome-relevant mechanisms independent of the extent of brain structural damage and activation of neuroglia. Possibly, sex and age are critical factors which should be considered in future longitudinal studies of the histopathological and behavioral outcomes in combined injury models.

## Figures and Tables

**Figure 1 biomedicines-12-01399-f001:**
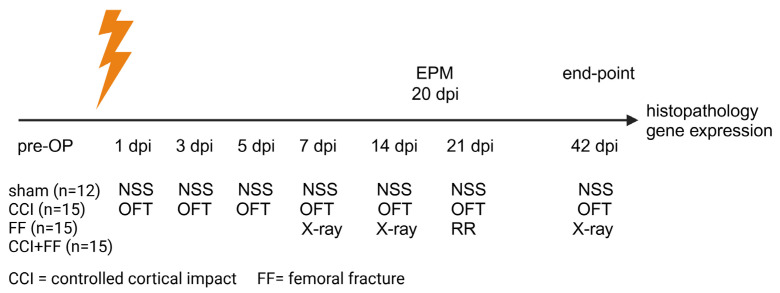
Experimental design and time line. Abbreviations: CCI = controlled cortical impact, EPM = elevated plus maze, FF = femoral fracture, NSS = Neurological severity score, OFT = open field test, RR = rotarod.

**Figure 2 biomedicines-12-01399-f002:**
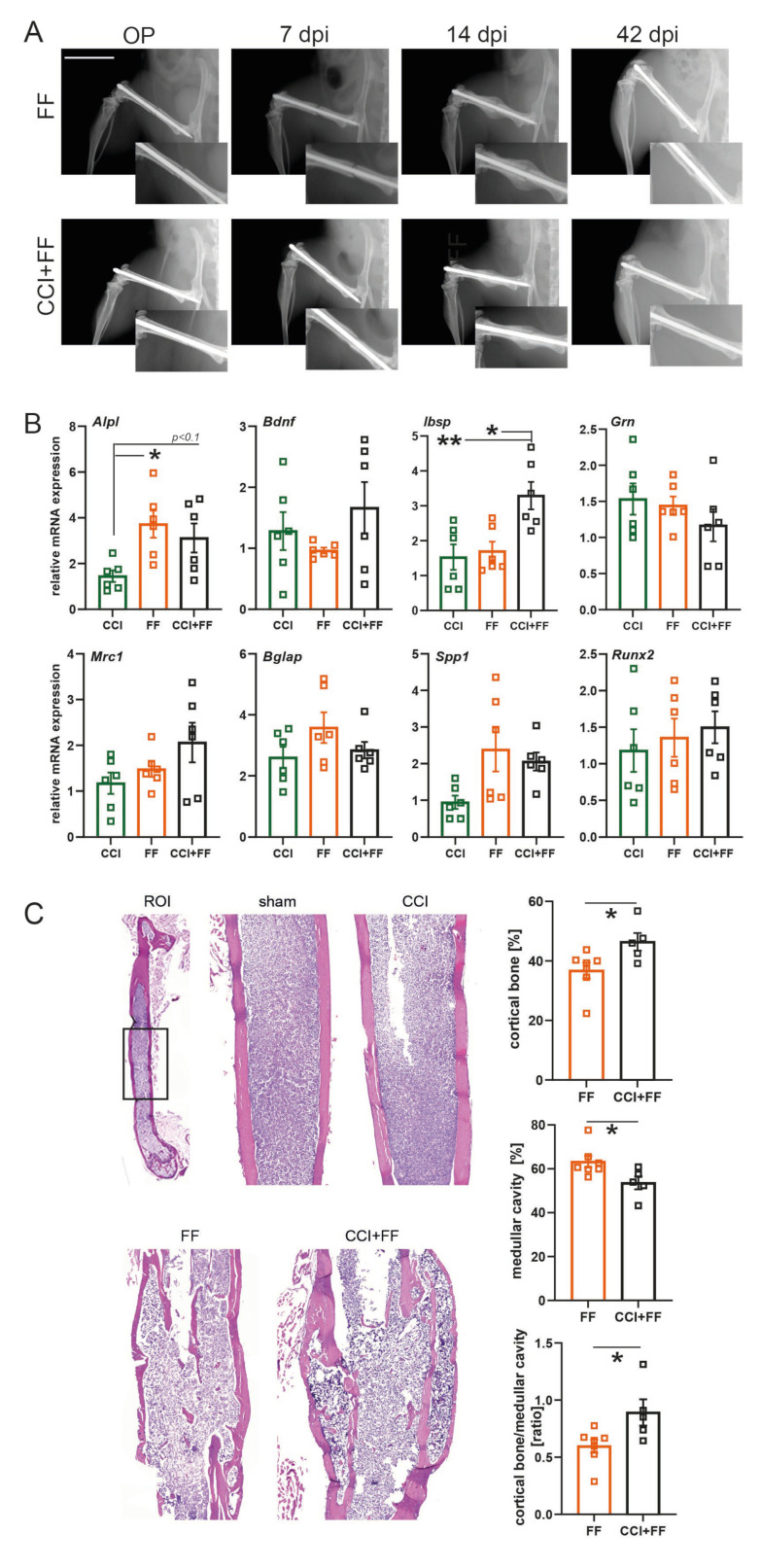
TBI up-regulates BSP gene expression in the fractured bone and promotes cortical bone formation (**A**) Representative X-ray images and detailed view of the fracture gaps of the osteosynthetic stabilized femoral fracture during surgical procedure as well as 7, 14, and 42 dpi. Scale bar: 1 cm. (**B**) Gene expression analyzes in the perifractural bone tissue normalized to GAPDH and to sham at 42 dpi (n = 6/group). The mRNA expression of alkaline phosphatase (Alp) was increased in the fractured bone of FF and CCI+FF mice compared to isolated CCI, while expression of bone sialoprotein was increased only in mice with combined injury compared to both isolated traumata. (**C**) HE staining of bone tissue 42 dpi revealed increased formation of cortical compared to trabecular bone in mice with combined injury compared to isolated FF. * *p* < 0.05, ** *p* < 0.01. Values of all data represent mean ± SEM; *p* values were calculated by Student’s *t*-test (**C**) or one-way ANOVA followed by Holm–Šidák’s multiple comparison test (**A**,**B**). CCI = controlled cortical impact, FF = femoral fracture, Alpl = alkaline phosphatase, Bdnf = brain-derived neurotrophic factor, Ibsp = integrin binding sialoprotein, Grn = granulin, Mrc1 = mannose receptor C-type 1, Bglap = bone gamma-carboxyglutamate protein, Spp1 = secreted phosphoprotein 1, Runx2 = RUNX family transcription factor 2.

**Figure 3 biomedicines-12-01399-f003:**
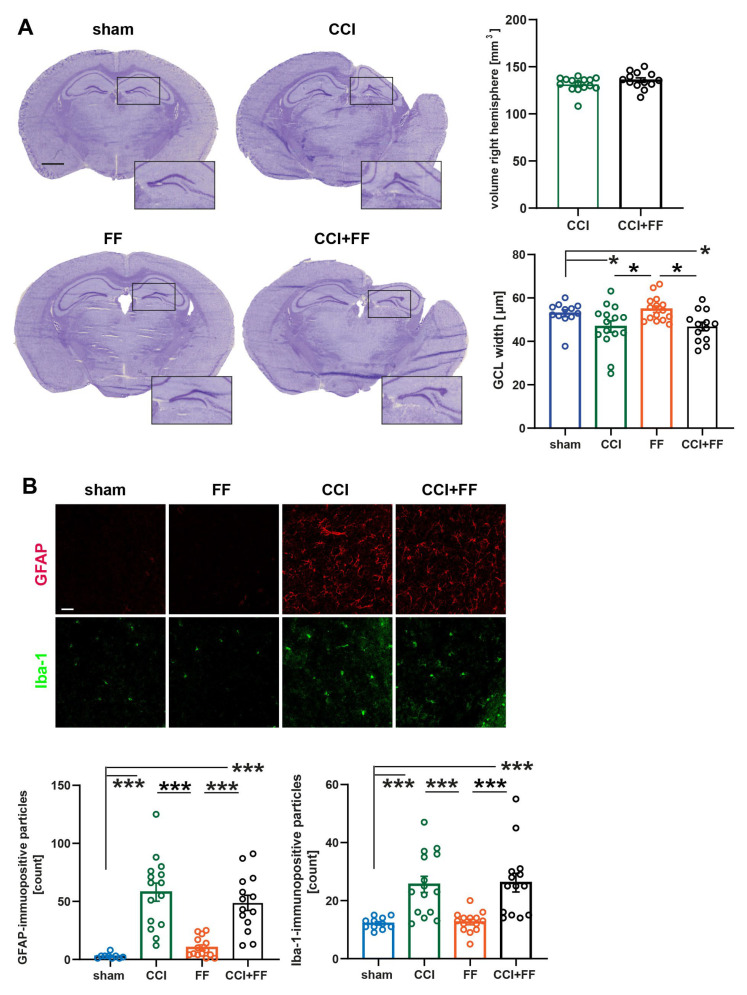
Femoral fracture does not aggravate long-term brain histopathology after TBI. (**A**) Representative images of cresyl violet-stained brain sections, Bregma −2.00 mm. Scale bar: 1 mm. Remaining volume of the right hemisphere was not affected by the additional femoral trauma 42 dpi. Outliers: CCI (volume right hemisphere) n = 1; sham, CCI (GCL width) each n = 1. (**B**) Representative immunofluorescence images of brain sections at 42 dpi showing microglia/macrophages (anti-Iba1, green) and reactive astrocytes (anti-GFAP, red) in the perilesional cortex of CCI and CCI+FF mice and corresponding regions of FF and sham mice (Bregma −2.00 mm, scale bar: 50 μm. Groups with cerebral trauma showed significantly increased counts of GFAP- and Iba-1-immunopositive particles in the perilesional region compared to sham and FF, but femoral fracture had no impact on these findings. Outliers: sham (GFAP) n = 2. Values of all data represent mean ± SEM; *p* values were calculated by Student’s *t*-test (**A**) one-way ANOVA (**A**,**B**) followed by Holm–Šidák’s multiple comparison test, * *p* < 0.05, *** *p* < 0.001. CCI = controlled cortical impact, FF = femoral fracture, GCL = granule cell layer, GFAP = glial fibrillary acidic protein, Iba1 = Ionized Calcium-Binding Adapter Molecule 1.

**Figure 4 biomedicines-12-01399-f004:**
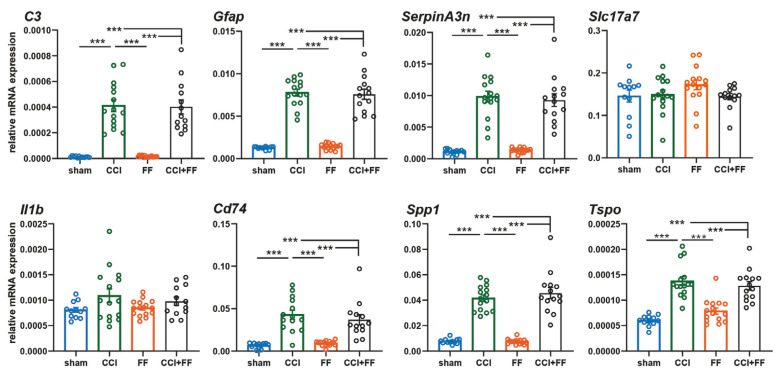
Femoral fracture has no influence on late TBI-induced gene expression of inflammation-associated markers Gene expression analyzes for brain inflammatory markers in perilesional brain tissue normalized to PPIA at 42 dpi. mRNA expressions of all SerpA3n, C3, GFAP, MHC2, TSPO, and SPP-1 were increased in CCI and CCI+FF mice compared to sham and FF, yet no significant differences could be detected between groups with cerebral injury. Outliers: CCI+FF (C3) n = 1; CCI, CCI+FF (CD47) each n = 1; CCI+FF (IL-1β) n = 1. *** *p* < 0.001. Values represent mean ± SEM; *p* values were calculated by one-way ANOVA and Holm–Šidák’s multiple comparison test. CCI = controlled cortical impact, FF = femoral fracture, C3 = complement C3, GFAP = glial fibrillary acidic protein, SerpinA3n = Serine protease inhibitor A3N, Slc17a7 = solute carrier family 17 member 7, Il1b = Interleukin 1b, Cd74 = cluster of differentiation 74, Spp1 = secreted phosphoprotein 1, Tspo = translocator protein.

**Figure 5 biomedicines-12-01399-f005:**
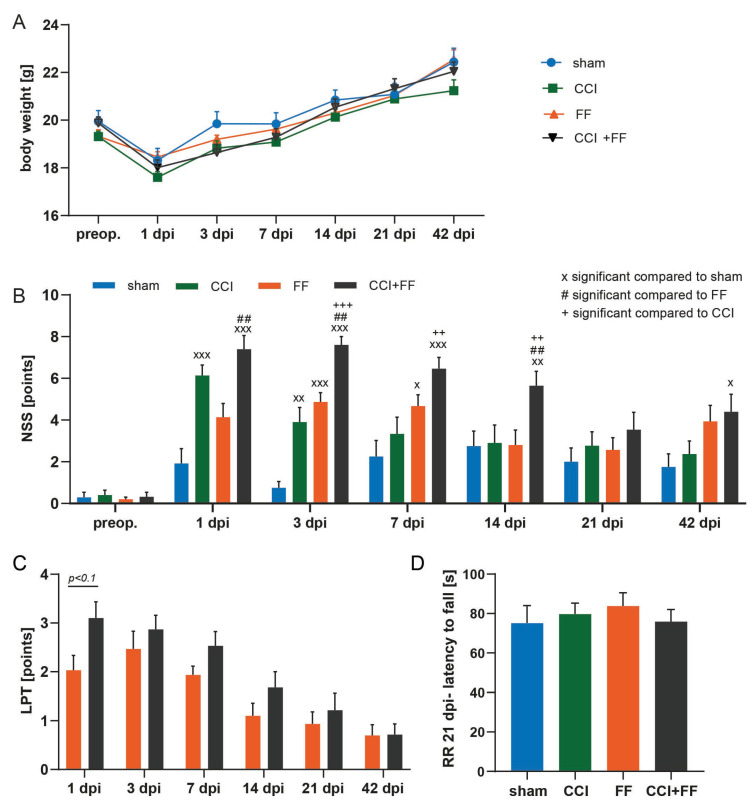
Concomitant TBI and femoral fracture prolongs neuromotor recovery. (**A**) Body weight remained comparable among all study groups at each time-point measured. Mice of all groups except FF suffered from a significant weight loss at 1 dpi compared to their pre-traumatic status (*p* < 0.5, not depicted in the graph), but regained their body weight quickly until 3 dpi. (**B**) NSS one day before as well as 1, 3, 7, 14, 21, and 42 dpi. Mice of the CCI group showed higher scores compared to sham until 3 dpi and had reached non-significantly increased values in comparison to their pre-traumatic baseline at 14 dpi. Animals with isolated FF exhibited increased scores compared to sham until 7 dpi and regained a non-significant level from baseline at 21 dpi. NSS of mice with combined injury (CCI+FF) were significantly higher compared to all other groups until 14 dpi and to sham until 42 dpi and remained increased in comparison to the pre-traumatic baseline until 42 dpi. x significant compared to sham (x *p* < 0.05, xx *p* < 0.01, xxx *p* < 0.001), # significant compared to FF ## *p* < 0.01), + significant compared to CCI (++ *p* < 0.01, +++ *p* < 0.001). (**C**) Mice with combined injury showed slightly increased impairment in lower limb mobility and strength in the leg performance test (LPT) at 1 dpi, yet this effect was no longer detectable at 3 dpi. (**D**) RR performance, quantified by latency to fall from the rotating wheel did not vary significantly among all study groups irrespective of their specific trauma at 21 dpi. Values of all data represent mean ± SEM; p values were calculated by one-way (**D**) or two-way ANOVA (**A**–**C**) followed by Holm–Šidák’s multiple comparison test. CCI = controlled cortical impact, FF = femoral fracture.

**Figure 6 biomedicines-12-01399-f006:**
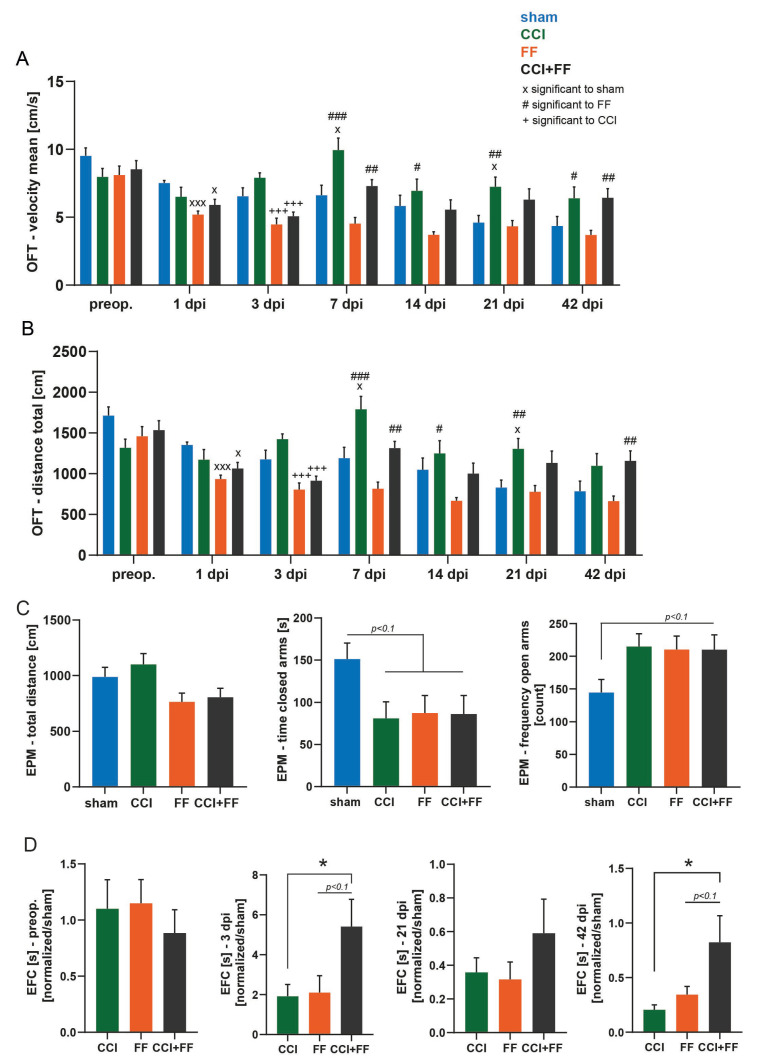
Concomitant TBI and femoral fracture affect anxiety-related behavioral patterns. (**A**,**B**) Mice with isolated FF and combined injury of CCI+FF moved less and slower in the OFT at 1 dpi in comparison to sham and until 3 dpi in comparison to CCI, quantified by video-tracking of mean velocity and total distance travelled in each 3 min recording episode. From 7 dpi until study’s endpoint an increased activity was visible in CCI and CCI+FF mice, while animals of sham and FF group rather tended to show less explorative ambitions. x significant compared to sham (x *p* < 0.05, xxx *p* < 0.001), # significant compared to FF (# *p* < 0.05, ## *p* < 0.01, ### *p* < 0.001), + significant compared to CCI (+++ *p* < 0.001). (**C**) All groups travelled comparable distances in the EPM test during the 5 min recording episode. Time spent in the closed arms as well as frequency of the open arms entries tended to be reduced respectively increased in all groups with surgical trauma in comparison to sham. (**D**) CCI+FF mice required prolonged duration to exit from center (EFC) in the OFT compared to isolated CCI or FF until 42 dpi. Outliers: FF (3 dpi) n = 2; CCI, FF, CCI+FF (21 dpi) each n = 1; CCI (42 dpi) n = 4, FF, CCI+FF (42 dpi) each n = 1, * *p* < 0.05. Values of all data represent mean ± SEM; p values were calculated by one-way (**C**,**D**) or two-way ANOVA (**A**,**B**) followed by Holm–Šidák’s multiple comparison test. CCI = controlled cortical impact, FF = femoral fracture, OFT = open field test, EPM = elevated plus maze, EFC = exit from center.

**Table 1 biomedicines-12-01399-t001:** Gene name, amplicon size, and oligonucleotide sequences (brain).

Gene Name, (Amplicon Size, bp)	Oligonucleotide Sequences 5′–3′ (fw: Forward, rev: Reverse)	Gene Bank Number
C3 (159)	fw-CCAGCTCCCCATTAGCTCTG rev-GCACTTGCCTCTTTAGGAAGTC	NM_009778.3
Gfap (120)	fw-CGGAGACGCATCACCTCTG rev-TGGAGGAGTCATTCGAGACAA	NM_001131020
Il1b (348)	fw-GTGCTGTCGGACCCATATGAG rev-CAGGAAGAAGGCTTGTGCTC	NM_008361
Cd74 (84)	fw-CCGCCTAGACAAGCTGACC rev-ACAGGTTTGGCAGATTTCGGA	NM_001042605
Ppia (146)	fw-GCGTCTSCTTCGAGCTGTT rev-RAAGTCACCCTGGCA	NM_008907
Tspo (152)	fw-GCCTACTTTGTACGTGGCGAG rev-CCTCCCAGCTCTTTCCAGAC	NM_009775
Serpina3n (167)	fw-GCCTCGTCAGGCCAAAAAG rev-TGAACGTGTCAAGAGGGTCAA	NM_009252
Spp1/Opn (151)	fw-ATGTCATCCCTGTTGCCCAG rev-GACTGATCGGCACTCTCCTG	NM_001204201.1
Slc17a7 (191)	fw-CCAACAGGGTCTTTGGCTTTG rev-CAGCCGACTCCGTTCTAAGG	NM_182993

**Table 2 biomedicines-12-01399-t002:** Gene name, amplicon size, and oligonucleotide sequences (bone).

Gene Name, (Amplicon Size, bp)	Oligonucleotide Sequences 5′–3′ (fw: Forward, rev: Reverse)	Gene Bank Number
Alpl	fw-TCAGGATGAGACTCCCAGGA rev-CACCCCGCTATTCCAAACAG	Nm_007431.3
Bdnf (187)	fw-TGCGGATATTGCGAAGGGTT rev-CACCTGGTGGAACATTGTGG	NM_007540.4
Bglap	fw-GAACAGACAAGTCCCACACAGC rev-TCAGCACAGTGAGCAGAAAGAT	NM_007541.3
Ibsp (158)	fw-GGACTGCCGAAAGGAAGGTT rev-GGCCGGTACTTAAAGACCCC	NM_008318.3
Gapdh (171)	fw-ACCCAGAAGACTGTGGATGG rev-CACATTGGGGGTAGGAACAC	NM_001289726.2
Grn (194)	fw-CTGCCCGTTCTCTAAGGGTG rev-ATCCCCACGAACCATCAACC	NM_008175.5
Mrc1 (120)	fw-GTGGAGTGATGGAACCCCAG rev-CTGTCCGCCCAGTATCCATC	NM_008625.2
Cd74 (84)	fw-CCGCCTAGACAAGCTGACC rev-AACAGGTTTGGCAGATTTCGGA	NM_001042605
Runx2 (213)	fw-CCTCGCTCTCTGTTCCTTCT rev-CATCTGCGTCTACTTGGTGC	NM_001146038.3
Spp1 (117)	fw-CCAGCCAAGGACTAACTACGA rev-AAAGCTTCTCCTCTGAGCTGC	NM_001204203.1

## Data Availability

The datasets generated and analysed during the current study are included in this published article or available from the corresponding author on reasonable request.

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
