# Peer review of "Brain–Bone Crosstalk in a Murine Polytrauma Model Promotes Bone Remodeling but Impairs Neuromotor Recovery and Anxiety-Related Behavior"

_biomedicines, 2024, doi:10.3390/biomedicines12071399_

Round 1

Reviewer 1 Report

Comments and Suggestions for Authors

1)              Title:  Brain-bone crosstalk in a murine polytrauma model influences callus formation, neuromotor recovery and long-term anxiety-related behaviour.

The title is generic, ‘influences’ is too neutral, and the title does not provide the specific findings of the present work.

Please, reconsider, as per the conclusions in the abstract:
- Results support previous findings on enhanced bone healing processes in
association with TBI.
- Behavioural assessments demonstrated an overall impaired recovery of neuromotor function and persistent abnormalities in anxiety-related behaviour in polytrauma mice.
- But results did not show remarkable neuropathological effects of bone fracture [Bone fracture did not aggravate pathological aspects (neuropathology or neuroinflammation assessed by cerebral lesion size, hippocampal integrity, astrocyte and microglia activation and gene expression],  suggesting the existence of outcome-relevant mechanisms independent of the extent of brain structural
damage and neuroinflammation.

But more importantly, the conclusion section is very clear targeting the finding: “In conclusion, we found that the late increase in BSP gene expression is associated with accelerated bone healing after combined TBI and bone fracture, reinforcing the role of bone trauma as a risk factor of long-lasting cognitive impairment.”

2) Lines 15-16: Brain and bone tissues were processed at 42 days after trauma and neuromotor impairment was assessed by neurological severity score, rotarod test, open field test, and elevated 17 plus maze test.

Please, check the order of methodological procedures, as behavior assessment in alive animals. Check also in the methods, as surgical procedures are listed after behavior.

3)The research work was performed in females. Please, provide an argument of the choice of only one sex, this one in particular and discuss expected results if two sexes were used instead.

4)The age of animals has an influence on the impact of lessions and subsequent effects. Please, justify the choice of 8-9 w.o. and discuss age effect as per literature available at other ages.

5)Repeated assessment of motor function can be also understood as a training rehabilitation process (i.e. https://doi.org/10.3390/biomedicines10050973 ) Please, discuss it with regards to the methodological procedures chosen in this respect (lines 85-86).

6)Please, add the methodological description of all the tests (RR and EPM are missing)

7)Why RR and EPM were only assessed at 21 dpi and not 42? To which extend this can be generalized and related to pathological results at end-point?

8)Please, add a scheme with the experimental design and timeline.

9)Number of outliers should be indicated.

10)Lines 228-229. Please, delete the guidelines “This section may be divided by subheadings. It should provide a concise and precise 228 description of the experimental results, their interpretation, as well as the experimental 229 conclusions that can be drawn.

3.1. Subsection

11)EFC exit for circle test appears in the results, but can’t be identified in the previous parts (i.e. abstract, methods). In fact, ‘exit from the center’ is a classical anxiety-related variable in the OF, not a test. In results, it is referred as a task. Please, correct, accordingly.

12)Figure 4.B The comparative patterns between groups seem consistent through the different days (progression from sham < cci < ff < cci+ff), but are different of 1 dpi (sham < ff < cci < cci+ff). Please, discuss.

13)Why the OF only lasted 3 minutes and not the 5 min standard for a 40x40cm2 or 50x50cm2 arena?

14) The conclusion (“we found that the late increase in BSP gene expression is associated with accelerated bone healing after combined TBI and bone fracture, reinforcing the role of bone trauma as a risk factor of long-lasting cognitive impairment”) refers to long-lasting cognitive impairment, but none of the behavioral assessments chosen in this experimental work was specific for cognitive function but motor and anxiety (despite exploration is always part of a test, OF and EPM are anxiety tests, not cognitive tests). Please, correct accordingly.

Author Response

REVIEWER 1:

Comments and Suggestions for Authors

1)              Title:  Brain-bone crosstalk in a murine polytrauma model influences callus formation, neuromotor recovery and long-term anxiety-related behaviour.

The title is generic, ‘influences’ is too neutral, and the title does not provide the specific findings of the present work.

Our response: We agree with this reviewer and changed the title to: Brain-bone crosstalk in a murine polytrauma model promotes bone remodeling but impairs neuromotor recovery and anxiety-related behavior

2) Lines 15-16: Brain and bone tissues were processed at 42 days after trauma and neuromotor impairment was assessed by neurological severity score, rotarod test, open field test, and elevated 17 plus maze test.

Please, check the order of methodological procedures, as behavior assessment in alive animals. Check also in the methods, as surgical procedures are listed after behavior.

Our response: The order of the methodological procedures in the abstract was changed as following: “Neuromotor and behavioral impairments were assessed by neurological severity score, open field test, rotarod test and elevated plus maze test. Brain and bone tissues were processed at 42 days after trauma.”

3) The research work was performed in females. Please, provide an argument of the choice of only one sex, this one in particular and discuss expected results if two sexes were used instead.

and

4) The age of animals has an influence on the impact of lessions and subsequent effects. Please, justify the choice of 8-9 w.o. and discuss age effect as per literature available at other ages.

Our response: We are grateful for this comment and addressed this point in the new subsection 4.3. “Limitations of this study” and write: This study has some limitations to be considered. First, we considered group housing important to avoid behavioral effects due to social isolation [62]. Therefore, only female mice were studied to enable group housing and avoid additional injuries caused by ranking battles, which often occur in group housing of male mice. However, sex-specific effects have been reported in patients as well as in mice after TBI [63-66] or bone fracture [67, 68]. Second, age is another outcome-relevant factor in experimental models of TBI [69, 70] or bone fracture [71, 72], which has not been considered in the present study using only 8-9 weeks-old mice. It would be therefore important to examine the influence of age and sex on the histopathological and behavioral outcomes in combined injury models. Third, behavioral testing using rotarod or the EPM was not performed at the end-point of this study at 42 dpi, thereby preventing conclusions on the persistence of the observed effects at 20 dpi or 21 dpi, respectively. Finally, additional tests to assess cognitive function would have provided a more comprehensive picture of changes in the behavioral spectrum in our combined injury model.

5) Repeated assessment of motor function can be also understood as a training rehabilitation process (i.e. https://doi.org/10.3390/biomedicines10050973 ) Please, discuss it with regards to the methodological procedures chosen in this respect (lines 85-86).

Our response: We thank this reviewer for pointing out this important aspect of repeated assessment. In the results subsection 3.4 we already mentioned habituation effects of behavioural testing and now added the following sentence to the discussion, subsection 4.2: However, it should be noted that repeated assessment of motor function potentially generates training effects and might be considered as a rehabilitation process [51]. To reduce this effect as a potential confounder, the assessment schedule remained identical among all subjects and focused on the performance in the NSS and OFT rather than repeated rotarod testing.

6) Please, add the methodological description of all the tests (RR and EPM are missing)

Our response: We apologize for this shortcoming in the methodological description and revised the respective part in the methods section.

7) Why RR and EPM were only assessed at 21 dpi and not 42? To which extend this can be generalized and related to pathological results at end-point?

Our response: Rotarod was tested only at 21 dpi as we observed in previous studies that CCI mice display at this time-point an impaired performance (Pöttker et al 2017, PMID: 28589257). However, we did not observe CCI-induced impairments in rotarod performance at 28 dpi (Hummel et al 2020, PMID: 32964418) and therefore decided to test only at 21 dpi. We added the reference Pöttker et al to the methods and write: Motoric behavioral impairment was further addressed by rotarod (RR) test at 21 dpi, a post-traumatic time point showing impaired RR performance in mice after CCI [34].

Regarding EPM testing, we considered the EPM to be more sensitive to handling effects as compared to other tests, e.g. OFT, as reported by Ueno et al 2020, PMID: 32103098, possibly affecting the interpretation of the results. However, we agree with this reviewer that single EPM testing at 42 dpi would have provided a clearer picture if brain-bone crosstalk effects on explorative and anxiety-like behaviour persist at the end-point of our study. We now address this issue in subsection 4.3. “Limitations of this study” and write: “Third, behavioral testing using rotarod or the EPM was not performed at the end-point of this study at 42 dpi, thereby preventing conclusions on the persistence of the observed effects at 20 dpi or 21 dpi, respectively. Finally, additional tests to assess cognitive function would have provided a more comprehensive picture of changes in the behavioral spectrum in our combined injury model”.

8) Please, add a scheme with the experimental design and timeline.

Our response: A scheme with experimental design and timeline was added as Fig. 1.

9) Number of outliers should be indicated.

Our response: Outliers were tested by ROUT Outlier test and numbers were indicated in the figure legends.

10) Lines 228-229. Please, delete the guidelines “This section may be divided by subheadings. It should provide a concise and precise 228 description of the experimental results, their interpretation, as well as the experimental 229 conclusions that can be drawn.

Our response: The remaining guideline instruction was removed from the text.

11) EFC exit for circle test appears in the results, but can’t be identified in the previous parts (i.e. abstract, methods). In fact, ‘exit from the center’ is a classical anxiety-related variable in the OF, not a test. In results, it is referred as a task. Please, correct, accordingly.

Our response: We thank the reviewer for this annotation, the term “task” was removed and the findings were addressed as “exit from center in the OFT” or “variable”.

12) Figure 4.B The comparative patterns between groups seem consistent through the different days (progression from sham < cci < ff < cci+ff), but are different of 1 dpi (sham < ff < cci < cci+ff). Please, discuss.

Our response: We thank the reviewer for highlighting this particular detail in the NSS and added it to the corresponding result section: “NSS of CCI mice was lower than in FF as a single injury during the experiment except for 1dpi, again underscoring the rapid recovery after isolated CCI compared to combined injury as well as the impact of the fracture impact on neuromotor scoring.”

13)Why the OF only lasted 3 minutes and not the 5 min standard for a 40x40cm2 or 50x50cm2 arena?

Our response: In the present study, data from the OFT were extracted from 3 minutes video-recording episodes. “Mean velocity” and “total distance” were used as parameters to assess general mobility in the OFT and therefore provide information about motoric dysfunction. A longer recording time could lead to effects of exhaustion and hereby reveal possible motoric impairment which remains hidden in the short episode and we will strongly consider to extend the OF time in upcoming experiments. The “exit from center” is a variable linked to the beginning of the OF. None of the animals required more than 3 minutes to leave the center of the arena, so we consider this parameter unaffected by the shorter recording time.

14) The conclusion (“we found that the late increase in BSP gene expression is associated with accelerated bone healing after combined TBI and bone fracture, reinforcing the role of bone trauma as a risk factor of long-lasting cognitive impairment”) refers to long-lasting cognitive impairment, but none of the behavioral assessments chosen in this experimental work was specific for cognitive function but motor and anxiety (despite exploration is always part of a test, OF and EPM are anxiety tests, not cognitive tests). Please, correct accordingly.

Our response: We fully agree with the reviewer and changed the term “cognitive” to “behavioural”.

Reviewer 2 Report

Comments and Suggestions for Authors

Review for the manuscript

Brain-bone crosstalk in a murine polytrauma model influences callus formation, neuromotor recovery and long-term anxiety related behaviour.

Dear Editor,

Thank you for the invitation to review for Biomedicine. Please find below my comments and suggestions.

OVERALL COMMENTS

            In this manuscript, the authors investigated the consequences of long-term reciprocal brain-bone interactions and assessed the course of behavioral impairments. They used mice subjected to controlled cortical impact and/or femoral fracture (FF) or sham surgery. Brain and bone tissues were processed at 42 days after trauma, and behavioral assessments were conducted over 42 days. Overall, the findings indicated that controlled cortical impact+Femur Fracture polytrauma mice showed increased bone formation compared to FF mice and increased mRNA expression of bone sialoprotein. Bone fracture did not aggravate neuropathology or neuroinflammation assessed by cerebral lesion size, hippocampal integrity, astrocyte and microglia activation, and gene expression.

TITLE

            The current title is "Brain-bone crosstalk in a murine polytrauma model influences callus formation, neuromotor recovery and long-term anxiety related behaviour."

I suggest changing for:

Brain-bone crosstalk in a murine polytrauma model influences callus formation, neuromotor recovery, and long-term anxiety-related behavior."

ABSTRACT

This section is adequate.

KEY-WORDS

In this section, we can find as key-words:

 Traumatic brain injury; bone fracture; osteopathology; neuropathology; neuroinflammation; behaviour. I suggest including mice.

INTRODUCTION

In this section, I suggest:

-       The inclusion of newer references. Please check on PUBMED, references published in 2024.

-       The inclusion of the definitions of all abbreviations in the text. As examples, definitions of IL-1β or TNFα are missing.

METHODS

            This section is adequate; each topic is well described.

The study was performed with animals (mice) and obtained the approval of the Committee of the Landesuntersuchungsamt Rheinland-Pfalz 72 (23177-0/G17-1-062) and following institutional and the ARRIVE guidelines.

RESULTS

The results were well described, but need some improvements. Please find my suggestions below:

In Figure 1, the images are of bad quality (the photos are nice, but the graphics are not. Please see the B and C parts of this Figure.

Figure 2 is good. Please use the same quality for Figure 1. I believe this modification will not require much work.

At the end of Figure 1 and Figure 2, please define the abbreviations.

I believe that the authors could create a table to summarize the results. The authors could create a Table showing all the “treatments” separately, but all in the same table.

Figure 4 is good. Please remove the space between the Figure and the legend. Check all the figures in this matter. Moreover, please include the definitions of CCI and RF in the legend. Please check all the Figures regarding this matter.

Figure 5 is also of good quality. Please check the legend for all the abbreviations that were used.

DISCUSSION

This section is adequate, but I suggest including newer references.

CONCLUSIONS

            These sections are adequate. However, I suggest including the limitations of this review.

            Please include the strengths and the limitations of this study.

Comments on the Quality of English Language

Minor.

Author Response

TITLE

            The current title is "Brain-bone crosstalk in a murine polytrauma model influences callus formation, neuromotor recovery and long-term anxiety related behaviour."

I suggest changing for:

Brain-bone crosstalk in a murine polytrauma model influences callus formation, neuromotor recovery, and long-term anxiety-related behavior."

Our response: According to the suggestions from the academic editor and Reviewer 1, we have changed the title to Brain-bone crosstalk in a murine polytrauma model promotes bone remodeling but impairs neuromotor recovery and anxiety-related behavior

KEY-WORDS

In this section, we can find as key-words:

 Traumatic brain injury; bone fracture; osteopathology; neuropathology; neuroinflammation; behaviour. I suggest including mice.

 Our response: We included “mice” as a keyword.

INTRODUCTION

In this section, I suggest:

-       The inclusion of newer references. Please check on PUBMED, references published in 2024.

-       The inclusion of the definitions of all abbreviations in the text. As examples, definitions of IL-1β or TNFα are missing.

Our response: We revised the introduction and added new references including 4 refrences published between 2020-2024. IL-1β or TNFα, are now defined.

METHODS

            This section is adequate; each topic is well described.

The study was performed with animals (mice) and obtained the approval of the Committee of the Landesuntersuchungsamt Rheinland-Pfalz 72 (23177-0/G17-1-062) and following institutional and the ARRIVE guidelines.

Our response: This part was revised as suggested by the editorial office to avoid self-plagiarism.

RESULTS

The results were well described, but need some improvements. Please find my suggestions below:

In Figure 1, the images are of bad quality (the photos are nice, but the graphics are not. Please see the B and C parts of this Figure.

Figure 2 is good. Please use the same quality for Figure 1. I believe this modification will not require much work.

At the end of Figure 1 and Figure 2, please define the abbreviations.

Our response: Figure 1 was included in better quality. We now defined all abbreviations in the figure legends. In addition, abbreviations for CCI and FF are also defined in Fig. 1.

I believe that the authors could create a table to summarize the results. The authors could create a Table showing all the “treatments” separately, but all in the same table.

Our response: We have started to design a table but we have to admit that we feel that the experimental design with different posttraumatic observation time-points, four experimental groups and various experimental readouts (histology, immunofluorescence, gene expression, behavioral tasks) is too complex to provide a well-structured table to provide a clear overview of our results.

Figure 4 is good. Please remove the space between the Figure and the legend. Check all the figures in this matter.

Our response: to the best of our knowledge, this is preformatted by the journal

Moreover, please include the definitions of CCI and RF in the legend. Please check all the Figures regarding this matter.

Figure 5 is also of good quality. Please check the legend for all the abbreviations that were used.

Our response: please see above.

DISCUSSION

This section is adequate, but I suggest including newer references.

Our response. We revised the discussion/conlcuisons and included newer references.

CONCLUSIONS

            These sections are adequate. However, I suggest including the limitations of this review.

            Please include the strengths and the limitations of this study.

Our response: We added a new subsection, 4.3. Limitations of this study

Round 2

Reviewer 1 Report

Comments and Suggestions for Authors

I'd like to thanks the authors for their point-by-point answer to this reviewer, providing clear explanations to all the issues raised. The revised version of the Ms. shows an effort to include or expand the rationale for methodological aspects, as requested. In the discussion section, the authors have also identified the limitations and argumented them appropiately. The title is now more informative on the findings of this research work and its different dimension outputs studied (promotion of bone remodeling but impairment of neuromotor recovery and anxiety-related behavior).
Overall, from my part, I consider the revised Ms. is acceptable in the present form and I'm glad that it will contribute to solve the scarcity of literature in this important topic.

I just realized about a mistake in the methods, as the work was done in females, but the authors, after corrections of other methodological issues, have left a sentence that refers to male sex

Line 101

Adult male C57BL/6N mice

Author Response

We would like to thank the editors and reviewers for their careful comments and helpful criticism.

Comment 1: I just realized about a mistake in the methods, as the work was done in females, but the authors, after corrections of other methodological issues, have left a sentence that refers to male sex.

Our response: We apologize and corrected this mistake.